# Therapeutic Potential of CHCHD2 in Ischemia–Reperfusion Injury: Mechanistic Insights into Nrf2-Dependent Antioxidant Defense in HK2 Cells

**DOI:** 10.3390/ijms26136089

**Published:** 2025-06-25

**Authors:** Yajie Hao, Xiaoshuang Zhou

**Affiliations:** 1Department of Nephrology, The Fifth Clinical Medical College of Shanxi Medical University, Taiyuan 030001, China; yajiehao66@163.com; 2Medicinal Basic Research Innovation Center of Chronic Kidney Disease, Ministry of Education, Shanxi Medical University, Taiyuan 030001, China

**Keywords:** ATP-depletion and recovery, ischemia/reperfusion AKI, oxidative stress, CHCHD2

## Abstract

Acute kidney injury (AKI) resulting from ischemia/reperfusion (I/R) poses a significant clinical challenge due to its high mortality and complex pathophysiology. Here, the protective actions of Coiled-coil-helix-coiled-coil-helix domain containing 2 (CHCHD2) in carbonyl cyanide m-chlorophenyl hydrazone (CCCP)-induced adenosine triphosphate depletion and recovery (ATP-D/R) injury in human kidney-2 (HK2) cells are examined. During ATP-D/R, expression levels of CHCHD2 were significantly reduced. The overexpression of CHCHD2 substantially reduced the levels of ROS, lipid peroxidation, apoptosis, kidney injury molecule-1 (KIM-1), and neutrophil gelatinase-associated lipocalin (NGAL), whereas the knockdown of CHCHD2 exacerbated cellular injury. Mechanistic studies further demonstrated that overexpression of CHCHD2 restored Nrf2 expression under ATP-D/R conditions, facilitated its nuclear translocation, and upregulated the downstream antioxidant enzyme HO-1. In contrast, the knockdown of Nrf2 reduced the cytoprotective actions of CHCHD2. These findings indicate that CHCHD2 reduces cellular damage by enhancing antioxidant defenses and reducing apoptosis through activating the Nrf2 axis, underscoring its potential as a therapeutic target for AKI.

## 1. Introduction

Acute kidney injury (AKI) is linked with high mortality. Ischemia/reperfusion (I/R) can lead to renal damage, especially in contexts such as renal surgery, organ transplantation, trauma, or shock [1,2]. I/R injury not only leads to renal cell death but also triggers cascades of apoptotic and inflammatory responses, further worsening kidney damage [3,4,5]. Therefore, elucidating the mechanisms behind I/R injury and investigating potential therapeutic strategies are essential for reducing the incidence of AKI and enhancing clinical outcomes.

During the development of AKI, injury to renal epithelial cells, especially proximal tubular cells, is a critical factor because these cells are crucial in maintaining renal function and their impairment often results in severe renal dysfunction [6,7]. As tubular injury occurs, various cytokines, inflammatory mediators, and renal biomarkers are released, reflecting the extent of kidney damage. Among these, kidney injury molecule-1 (KIM-1) and neutrophil gelatinase-associated lipocalin (NGAL) are early indicators of AKI. Their levels are significantly elevated during I/R injury, making them sensitive indicators for monitoring renal damage [8].

The Human Kidney-2 (HK2) cell line, derived from normal human kidney proximal tubular epithelia with immortalization with human papillomavirus 16 (HPV16) E6/E7 gene transfection, retains the phenotype and functions of proximal tubular cells. HK2 cells are commonly used to model I/R-induced AKI [9]. Currently, several in vitro methods exist to simulate I/R injury, including the use of hypoxia incubators/chambers, chemical induction, and oxygen depletion techniques, which induce cellular hypoxia followed by recovery to mimic I/R conditions [10,11,12]. Among these, the adenosine triphosphate depletion and recovery (ATP-D/R) model is widely utilized to replicate the pathophysiological changes associated with tissue ischemia characterized by an interruption of energy supply leading to ATP depletion and the subsequent restoration of ATP during reperfusion, thereby providing insights into the mechanisms underlying I/R damage [13,14]. Carbonyl cyanide m-chlorophenyl hydrazone (CCCP), which uncouples mitochondria, exerts significant cytotoxic effects. Its mechanism involves acting as a protonophore that dissipates the mitochondrial proton gradient (membrane potential), thereby disrupting the inner mitochondrial membrane electrochemical gradient and inhibiting ATP synthesis [15]. Consequently, CCCP is extensively employed not only in studies investigating mitochondrial function, apoptosis, and autophagy [16,17] but also as a common agent to induce ATP-D/R in order to model in vitro I/R damage [18,19].

Within this context, Coiled-coil-helix-coiled-coil-helix domain containing 2 (CHCHD2), a key regulator of mitochondrial function, has recently garnered attention for its role in various cellular stress responses [20,21]. Research indicates that CHCHD2 not only can modulate mitochondrial function but also possesses potent antioxidant and anti-apoptotic properties [22,23,24]. Recent studies further suggest that CHCHD2 may regulate the Nrf2 axis [22]. Nrf2 is a crucial transcription factor that orchestrates the cellular antioxidant response; upon activation, it upregulates the expression of multiple antioxidant enzymes including HO-1, NAD(P)H quinone dehydrogenase 1 (NQO1), and glutamate-cysteine ligase catalytic and modifier subunits (GCLC/GCLM) to mitigate oxidative stress [25]. Overexpression of CHCHD2 has been shown to induce Nrf@ translocation to the nucleus, facilitating its binding to antioxidant response elements (AREs) and thereby enhancing the antioxidant capacity of the cell [22]. However, this mechanism has not yet been elucidated in the context of AKI. Considering that ATP-D/R injury is accompanied by substantial ROS generation and intense oxidative stress, which ultimately leads to apoptosis and necrosis [26,27], CHCHD2 may exert a protective effect during ATP-D/R injury.

Accordingly, here, the influence of CHCHD2 on HK2 cell function under ATP-D/R conditions and its potential protective mechanism through the modulation of the Nrf2 axis were investigated. Thus, by potentially alleviating oxidative stress and inhibiting apoptosis, CHCHD2 can be utilized in treating renal injury in AKI. The ultimate goal is to delineate the role of CHCHD2 in protecting renal cells from the severe effects of ischemia and reperfusion, thereby contributing to the development of targeted interventions that could improve outcomes for patients suffering from AKI.

## 2. Results

### 2.1. CCCP Toxicity Experiment and ATP-D/R Oxidative Stress Assessment

CCCP uncouples mitochondria by disruption of the proton gradient across the inner membrane, which leads to diminished ATP synthesis and impaired mitochondrial respiration, thereby inducing a hypoxic effect on cells [28,29]. In our experiments, cells were exposed to varying concentrations of CCCP for 4 h to induce hypoxia, followed by a 2 h recovery period. The results showed a significant decline in cell viability with increasing CCCP concentrations (Figure 1A). Using the Cell Counting Kit-8 (CCK8) assay, a concentration of 20 μM was determined to be optimal for modeling ATP depletion and recovery (ATP-D/R) injury. Additionally, ATP-D/R treatment led to a significant rise in intracellular contents of malondialdehyde (MDA), indicating increased lipid peroxidation damage, while the concentration of glutathione (GSH) and activity of superoxide dismutase (SOD) were markedly reduced (Figure 1B–D), indicating a high level of oxidative stress leading to cellular injury.

### 2.2. Effects of CHCHD2 Overexpression on Cellular Oxidative Stress Markers

We first assessed CHCHD2 expression under ATP-D/R conditions using Western blot analysis, which revealed a significant reduction in its expression (Figure 2A). This observation led us to hypothesize that overexpressing CHCHD2 might alleviate the oxidative stress damage induced by ATP-D/R. To investigate this, we created CHCHD2 overexpressing (OE) cell lines and compared them with cells transfected with an empty vector (EV) as controls. Successful overexpression was verified by Western blot (WB) (Figure 2B). Subsequently, we exposed these cells to CCCP-induced ATP-D/R conditions and assessed various oxidative stress markers and cell viability indicators. Notably, cells with CHCHD2 overexpression demonstrated significant protective effects: there was a considerable reduction in intracellular levels of MDA compared to the ATP-D/R EV group, indicating reduced lipid peroxidation damage. Additionally, levels of GSH, activity of SOD, and overall cell viability were markedly greater in the CHCHD2 OE group relative to the ATP-D/R EV group (Figure 2C–F), indicating that CHCHD2 overexpression could enhance cellular antioxidant defenses. Moreover, JC-1 analysis further demonstrated that CHCHD2 OE attenuated ATP-D/R-induced mitochondrial membrane potential loss, indicating its protective role against mitochondrial injury(Figure 2G).

### 2.3. CHCHD2 Overexpression Mitigated Apoptosis and Cellular Injury

Additionally, we assessed markers of apoptosis and cellular injury. Flow cytometry analysis demonstrated that the apoptosis rate was markedly elevated in the ATP-D/R EV group; however, markedly reduced apoptosis was found following CHCHD2 overexpression (Figure 3A,B), underscoring its anti-apoptotic effect. Furthermore, quantitative PCR analysis revealed that RNA levels of KIM-1 and NGAL, which are biomarkers for AKI, were significantly elevated in the ATP-D/R EV group but were substantially reduced following CHCHD2 OE (Figure 3C,D). Moreover, CHCHD2 OE also led to a reduction in NGAL protein levels (Figure 3E). Concurrently, intracellular ROS levels, measured using a fluorescence probe, were decreased in the CHCHD2 OE group relative to the ATP-D/R group (Figure 3F). These findings offer strong evidence that CHCHD2 overexpression can confer protection against ATP-D/R injury by mitigating oxidative stress, inhibiting apoptosis, and reducing the expression of AKI biomarkers.

### 2.4. CHCHD2 Knockdown Exacerbated Cellular Injury

To further explore the role of CHCHD2 in cellular function, we subjected cells with stable CHCHD2 knockdown (KD) to ATP-D/R treatment. Cells transfected with non-targeting short hairpin RNA (shNC) served as controls. Initially, we confirmed the efficiency of the KD (Figure 4A). Upon ATP-D/R intervention, cells with CHCHD2 KD displayed markedly elevated oxidative stress and injury markers relative to the shNC group. Specifically, markedly raised MDA levels and reduced GSH and SOD activities were seen in the knockdown group (Figure 4B–D), indicating a more pronounced impairment of cellular antioxidant capacity. Additionally, the CCK8 assay showed that the viability of CHCHD2 KD cells was significantly less than in shNC cells, suggesting a reduced tolerance to ATP-D/R-induced injury (Figure 4E). Furthermore, under ATP-D/R conditions, RNA expression levels of NGAL and KIM-1 in the CHCHD2 KD cells were elevated relative to those in shNC cells, indicating an exacerbation of cell injury (Figure 4F,G). Finally, CHCHD2 KD further increased NGAL protein levels under ATP-D/R, suggesting a detrimental role of CHCHD2 deficiency (Figure 4H). Although the difference in KIM-1 expression was non-significant, likely due to high variability within the group, these results demonstrate that CHCHD2 KD exacerbates cell injury under ATP-D/R conditions, emphasizing the critical role of CHCHD2 in mitigating cellular damage.

### 2.5. Mechanistic Investigation of CHCHD2 Regulation of the Nrf2 Axis

Nrf2 is responsible for regulating the transcription of antioxidant genes, and its activation significantly enhances the production of downstream antioxidants, including HO-1 and NQO1, thus reducing oxidative stress. To investigate the molecular mechanism behind the protective effects of CHCHD2, we examined its interaction with the ARE axis. Our experimental results showed that under ATP-D/R conditions, Nrf2 protein expression in HK2 cells was significantly reduced, however, CHCHD2 OE partially restored Nrf2 levels (Figure 5A). Additionally, analysis of nuclear protein extracts showed that Nrf2 levels were higher in CHCHD2 OE cells relative to the controls, indicating that CHCHD2 promotes Nrf2 transfer to the nucleus, thereby increasing its interaction with ARE and enhancing the cell’s ability to mitigate oxidative stress (Figure 5B). Moreover, levels of HO-1, downstream of Nrf2, were found to be upregulated in the CHCHD2 OE group (Figure 5C), further indicating that CHCHD2 OE can enhance the activity of the Nrf2-ARE pathway. To confirm whether CHCHD2 protection against ATP-D/R injury relies on the Nrf2 pathway, we used shRNA to KD Nrf2 expression in CHCHD2 OE cells (Figure 5D). We found that under ATP-D/R conditions, Nrf2 KD significantly reduced the antioxidative stress protective effect provided by CHCHD2 OE: malondialdehyde (MDA) levels increased, whereas GSH and SOD activities decreased, and cell viability was substantially reduced (Figure 5E–G). These findings underscore the essential role of Nrf2 in CHCHD2-mediated cytoprotection.

## 3. Discussion

CHCHD2, a mitochondrial protein, has attracted significant attention recently. While the involvement of CHCHD2 in mitochondrial energy metabolism and respiratory chain regulation is well-known, it has also been proposed to protect against intracellular stress [30]. A number of prior studies suggest that under oxidative stress, CHCHD2 can influence both apoptotic and antioxidant defense pathways [22,24,31]. ATP-D/R injury, closely linked with oxidative stress, is characterized by a surge of ROS during the recovery phase. This increase in ROS can overwhelm cellular antioxidant systems like GSH and SOD, leading to damage to lipids, proteins, and DNA and ultimately resulting in cellular dysfunction or apoptosis [32,33]. Based on this background, this study evaluated the ability of CHCHD2 to protect against CCCP-induced ATP-D/R damage in HK2 cells as well as the mechanisms involved.

In this study, marked downregulation of CHCHD2 expression was observed in the CCCP-mediated ATP ATP-D/R injury HK2 cell model. This finding contrasts with previous reports of CHCHD2 upregulation in SH-SY5Y neuroblastoma cells treated with tert-butyl hydroperoxide (TBHP) or hydrogen peroxide (H_2_O_2_) [22]. We hypothesize that this discrepancy arises from fundamental differences in the modes of stress. Specifically, unlike TBHP and H_2_O_2_, CCCP-induced ATP-D/R not only provokes a ROS burst but also lowers the mitochondrial membrane potential, resulting in more severe mitochondrial damage and activation of the proteasomal and autophagic degradation pathways. This enhanced proteolytic activity may accelerate CHCHD2 turnover, leading to its reduced expression under ATP-D/R conditions.

Consistent with the induction of oxidative stress, ATP-D/R was found to significantly increase the levels of MDA while reducing the activity of SOD, the GSH content, and overall cell viability in HK2 cells. To examine whether CHCHD2 could mitigate these injuries, we overexpressed CHCHD2 in the ATP-D/R model. This showed a pronounced attenuation of cellular damage, accompanied by an inhibition of apoptosis, improvements in the levels of oxidative stress markers, and increased cell survival. These results closely mirror those obtained in TBHP-treated SH-SY5Y cells, where CHCHD2 OE similarly reduced mitochondrial and intracellular ROS and suppressed the expression of cleaved Caspase-3, thereby lowering apoptosis rates [22]. Conversely, CHCHD2 KD in the present ATP-D/R model exacerbated both oxidative stress and cell injury, a phenotype that aligns with Drosophila studies showing that CHCHD2 deficiency (dCHCHD2^−^/^−^ or dCHCHD2^H43 mutants) led to reduced survival following H_2_O_2_ challenge, increased in vivo levels of 4-hydroxy-2-nonenal (4-HNE) and 8-hydroxy-2′-deoxyguanosine (8-OHdG), and a lowered GSH/GSSG ratio, collectively indicative of increased oxidative damage [24]. Moreover, CHCHD2 OE significantly reduced the upregulation of the AKI biomarkers KIM-1 and NGAL under ATP-D/R, whereas CHCHD2 KD increased their expression. Together, these data confirm previous findings at the molecular level and establish CHCHD2 as a critical regulator of redox homeostasis and cell survival.

Mechanistically, earlier dual-luciferase reporter assays in SH-SY5Y cells demonstrated that CHCHD2 overexpression enhanced the activity of ARE-luciferase, indicating activation of the Nrf2-ARE pathway [22]. Extending these insights to HK2 cells, we confirmed that CHCHD2 promoted nuclear translocation of Nrf2 in HK2 cells, thereby upregulating the expression of downstream antioxidant genes. Importantly, siRNA-mediated knockdown of Nrf2 abrogated the protective effects of CHCHD2, namely the reduction in MDA levels and the restoration of GSH and SOD activities under ATP-D/R, demonstrating that the cytoprotective function of CHCHD2 is dependent on Nrf2.

It should be noted that while CCCP-induced ATP-D/R provides a rapid and robust in vitro model of oxidative injury, it may not fully recapitulate the complex pathophysiology of in vivo ischemia–reperfusion. Future studies should therefore employ animal models of renal I/R injury and utilize in vivo delivery methods, such as adeno-associated virus vectors or nanoparticle systems, to induce overexpression of CHCHD2 in the kidney. Such investigations would be crucial to verify the protective role of CHCHD2 in vivo and assess its therapeutic potential. Finally, whether CHCHD2 interacts directly with Nrf2 or modulates its stability and translocation via upstream signaling intermediates remains an important avenue for further exploration.

## 4. Materials and Methods

### 4.1. Chemicals and Reagents

CCCP (HY-100941, ≥99% purity) was obtained from MedChemExpress (Monmouth Junction, NJ, USA). MEM (PIG0029) was sourced from BOSTER (Wuhan, China), and glucose-free MEM (PM150443) was purchased from Procell (Wuhan, China). Fetal bovine serum (SA211.02) was acquired from Cell Max (Beijing, China), and the Penicillin-Streptomycin-Amphotericin B Solution (P7630) was obtained from Solarbio (Beijing, China). Assay kits for superoxide dismutase (SOD, S0101S), cell counting kit-8 (CCK8, C0037), malondialdehyde (MDA, S0131S), and glutathione (GSH, S0053) were provided by Beyotime Biotechnology (Shanghai, China). The ROS detection kit (MA0219-1) was procured from Meilunbio (Dalian, China). Primary antibodies, including anti-CHCHD2 antibody (1:1000, 19424-1-AP) and β-actin monoclonal antibody (1:10,000, 66009-1-Ig), were purchased from Proteintech (Wuhan, China). Additionally, antibodies for Nrf2 (1:1000, 12721T) and FLAG (1:1000, 14793T) were obtained from Cell Signaling Technology (Danvers, MA, USA). The HRP-conjugated AffiniPure Rabbit Anti-goat IgG (H + L) secondary antibody (1:10,000, SA00001-2; BA1060) was obtained from BOSTER (Wuhan, China). The primary antibody against Histone H3 (1:1000, CY6587) was sourced from Abways (Shanghai, China). The M5 HiPer Universal RNA Mini Kit (MF036-01), M5 Super Plus qPCR RT Kit with gDNA Remover (MF166-plus-T), and 2X M5 HiPer SYBR Premix EsTaq (MF787-T) were all purchased from Mei5bio (Beijing, China).

### 4.2. Cell Culture and Treatment

HK2 cells were revived from liquid nitrogen storage and subsequently grown in MEM enriched with 10% FBS and 1% penicillin-streptomycin-amphotericin B. The cells were kept in a humidified incubator at 37 °C with 5% CO_2_, with replacement of media at 2–3 day intervals, and cell morphology was regularly checked.

For the ATP-D/R model in HK2 cells, we used 20 μM CCCP and replaced the medium with glucose-free MEM. The cells were kept for 4 h in a standard incubator to induce hypoxia, followed by a medium swap to normal culture medium for an additional 2 h of incubation to facilitate recovery.

HK2 cells were also infected with either CHCHD2 overexpression lentivirus or an empty vector lentivirus provided by Shanghai Genechem Co., Ltd., Shanghai, China. After 48 h, stable transfectants were selected using 4 μg/mL puromycin for a duration of two weeks. In addition, stable shCHCHD2 and shControl HK2 cell lines were sourced from Applied Biological Materials in Vancouver, Canada. HK2 cells were further transfected with Nrf2-shRNA supplied by Shanghai Genechem Co., Ltd., China, to accomplish Nrf2 knockdown.

### 4.3. Analysis of Cell Viability

CCK-8 assays were performed as directed. Specifically, cells were inoculated in 96-well plates and grown overnight. Once they reached 70–80% confluence, the cells were subjected to the planned ATP-D/R treatment. After this, 10 μL/well of CCK-8 solution was introduced for 1 h at 37 °C, followed by the reading of absorbance at 450 nm with a microplate reader.

### 4.4. Reactive Oxygen Assays

In this study, we utilized HK2 cells to determine intracellular ROS levels using a DCFH-DA assay kit. When the cells reached 70% confluence, the existing medium was discarded and replaced with a DCFH-DA working solution at a 1:1000 dilution to achieve a final concentration of 10 μM. The cells were then incubated at 37 °C in the dark for 30 min. After the incubation, the cells underwent three rinses with serum-free medium, followed by the ATP-D/R treatment. ROS concentrations were then measured by capturing the fluorescence signals using a fluorescence microscope.

### 4.5. GSH Assay

Intracellular GSH levels were assessed using a GSH and GSSG assay kit (Beyotime Biotechnology; Product No. S0053), as directed. Briefly, after homogenizing the samples and deproteinizing them with the provided reagent, total glutathione (GSH plus GSSG) was quantified through an enzymatic recycling method using DTNB and glutathione reductase. The absorbance was measured at 412 nm. To specifically measure GSSG, endogenous GSH was selectively depleted. The GSH levels were then determined by subtracting twice the GSSG concentration from the total glutathione measured.

### 4.6. SOD Assay

Total SOD activity was determined using a WST-8-based assay kit (Beyotime; Product No. S0101), as directed. Briefly, cells were lysed using the SOD sample preparation solution in the kit, followed by centrifugation and retention of the supernatant. The assay involved mixing a designated volume of the sample with a freshly prepared WST-8/enzymatic working solution, followed by adding the reaction start solution. Following incubation (30 min, 37 °C), absorbances at 450 nm were read. SOD activity was determined by calculating the inhibition rate of formazan dye formation, with results normalized and expressed as units per mg of protein.

### 4.7. MDA Assay

Lipid peroxidation was examined by determining MDA contents using a kit (Beyotime Biotechnology; Product No. S0131). Briefly, HK2 cells were lysed, and the supernatant was collected. This sample was then combined with a working solution containing Thiobarbituric Acid (TBA) and an antioxidant reagent, followed by heating (100 °C, 15 min) to yield the MDA-TBA adduct. After the mixture was allowed to cool and was centrifuged, the absorbance of the red-colored adduct was measured at 535 nm using a microplate reader. MDA concentrations were then determined using a standard curve prepared with known MDA standards.

### 4.8. Nuclear Protein Extraction

Firstly, the nuclear protein extraction reagent was dissolved at room temperature, followed by placement on ice and the addition of PMSF to 1 mM. Next, adherent cells were rinsed with PBS, harvested via scraping or EDTA treatment, and centrifuged to remove the supernatant and obtain a cell pellet. Then, based on the cell count, an appropriate volume of extraction reagent was added to the pellet, mixed thoroughly, and subjected to vigorous vortexing for 15–30 s. The sample was immediately transferred to an ice bath and vortexing was repeated every 1–2 min for a total of 30 min to ensure complete cell lysis. Finally, the lysate was centrifuged (12,000–16,000× *g*, 10 min, 4 °C), with careful collection of the supernatant representing the purified nuclear protein extract.

### 4.9. WB

Following cell sample processing, protein extracts were quantified using BCA assays, and equal protein amounts were electrophoresed on SDS-PAGE, followed by transfer to PVDF membranes, which were then blocked (5% skim milk in TBST, 1 h, ambient temperature). The blots were treated overnight with primary antibodies at 4 °C, and after three washes with TBST, were probed with HRP-conjugated secondary antibodies (1 h, ambient temperature). After further washing, protein bands were visualized by chemiluminescence and imaged.

### 4.10. RNA Isolation and qRT-PCR

Total cellular RNA was extracted using a M5 HiPer Universal RNA Mini Kit (MF036-01) as directed, and its purity and concentration were assessed via spectrophotometry. cDNA was synthesized using the M5 Super Plus qPCR RT Kit with gDNA Remover (MF166-plus-T), as directed. Subsequently, qRT-PCR was conducted using gene-specific primers and the 2X M5 HiPer SYBR Premix EsTaq (MF787-T) in a SYBR Green detection system. Gene expression was normalized to the internal control and quantified using the 2^−ΔΔCt^ method. The sequences of primers used for qRT-PCR are listed in Table 1.

### 4.11. Statistical Analysis

Data are presented as mean ± standard error of the mean (SEM). Statistical analyses were conducted using GraphPad Prism 9 software. Comparisons between two groups were made using a two-tailed Student’s *t*-test, and differences among multiple groups were assessed using one-way analysis of variance (ANOVA) with Tukey’s post hoc test for further analysis. *p* < 0.05 was deemed statistically significant.

## 5. Conclusions

To summarize, the findings indicated that CHCHD2 protected against CCCP-induced ATP-D/R damage in HK2 cells. The protective mechanism primarily involved the mitigation of oxidative stress and reduced apoptosis, notably through the activation of the Nrf2 axis. By enhancing the levels of antioxidant enzymes, CHCHD2 effectively reduced ROS production and lipid peroxidation levels, thereby enhancing cell viability. These findings not only expanded our understanding of cellular stress response regulation but also provided important experimental evidence for the potential development of novel renal protective strategies. Future research should further elucidate the role of CHCHD2 under various pathological conditions and validate its protective effects in animal models and clinical trials, ultimately paving the way for its application in AKI.

## Figures and Tables

**Figure 1 ijms-26-06089-f001:**
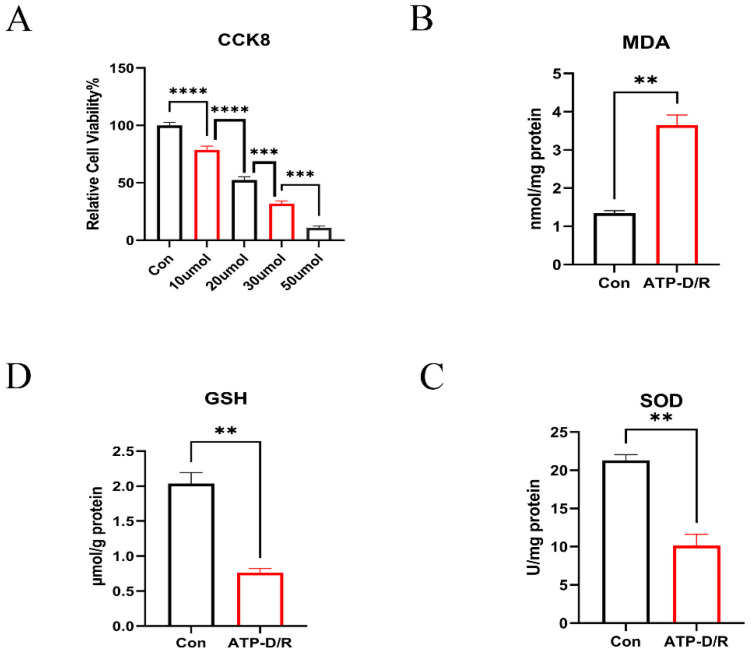
CCCP toxicity and ATP-D/R-induced oxidative stress changes. (**A**) HK2 cell viability was evaluated using the CCK8 assay after exposure to increasing concentrations of CCCP, demonstrating a notable decline in viability at higher CCCP levels. (**B**) Changes in intracellular MDA levels following ATP-D/R treatment. (**C**) Fluctuations in GSH levels post ATP-D/R treatment. (**D**) Changes in SOD activity after ATP-D/R treatment. Data are expressed as mean ± standard error of the mean (SEM) (n = 3). Statistical significance was determined using an unpaired Student’s *t*-test, with ns indicating no significance, ** *p* < 0.01, *** *p* < 0.001, and **** *p* < 0.0001.

**Figure 2 ijms-26-06089-f002:**
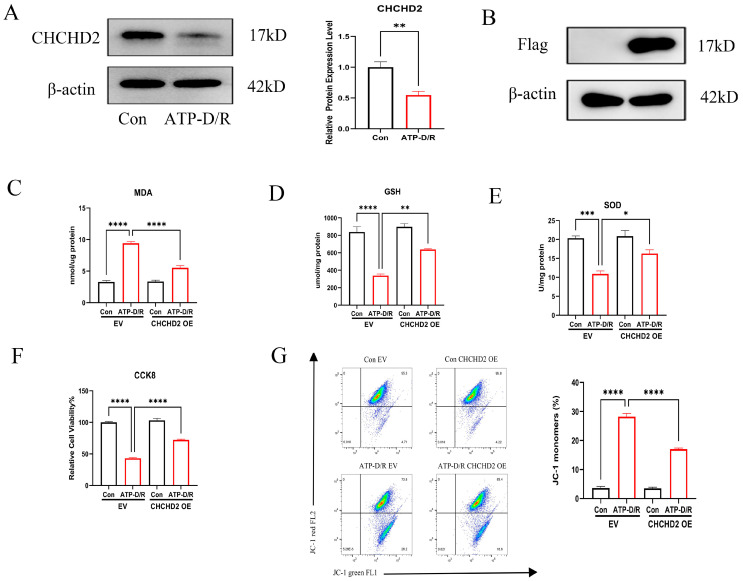
CHCHD2 overexpression alleviated ATP-D/R-induced oxidative stress injury. (**A**) Under ATP-D/R conditions, CHCHD2 expression was significantly reduced. (**B**–**F**) Validation of CHCHD2 overexpression and its protective effects: CHCHD2 overexpression significantly decreased MDA levels, restored GSH levels and SOD activity, and improved cell viability. (**G**) Overexpression of CHCHD2 mitigated mitochondrial injury induced by ATP-D/R. Values are presented as the mean ± standard error of the mean (SEM) (n = 3). Statistical significance was determined using a one-way ANOVA followed by Tukey’s multiple comparisons test, with ns indicating no significance, * *p* < 0.05, ** *p* < 0.01, *** *p* < 0.001, and **** *p* < 0.0001.

**Figure 3 ijms-26-06089-f003:**
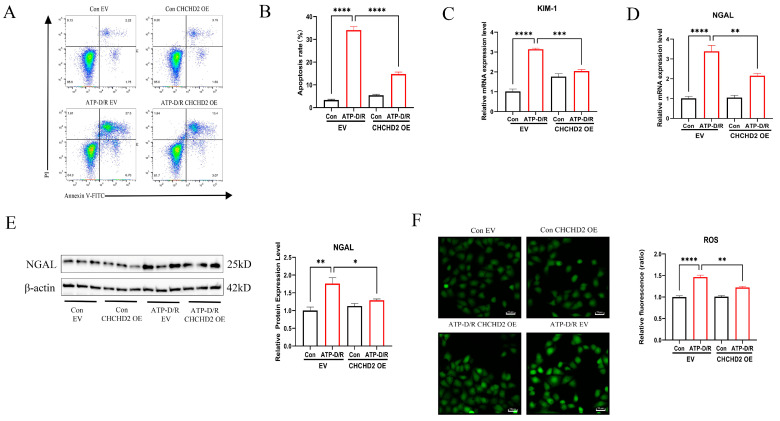
CHCHD2 overexpression attenuated apoptosis and injury. (**A**–**D**) CHCHD2 overexpression significantly reduced apoptosis levels and markedly decreased the RNA levels of kidney injury biomarkers KIM-1 and NGAL. (**E**) Overexpression of CHCHD2 attenuated the upregulation of NGAL protein expression induced by ATP-D/R. (**F**) Fluorescence probe analysis demonstrated that CHCHD2 overexpression significantly decreased intracellular ROS levels. Values are presented as the mean ± standard error of the mean (SEM) (n = 3). Statistical significance was determined using a one-way ANOVA followed by Tukey’s multiple comparisons test, with ns indicating no significance, * *p* < 0.05, ** *p* < 0.01, *** *p* < 0.001, and **** *p* < 0.0001.

**Figure 4 ijms-26-06089-f004:**
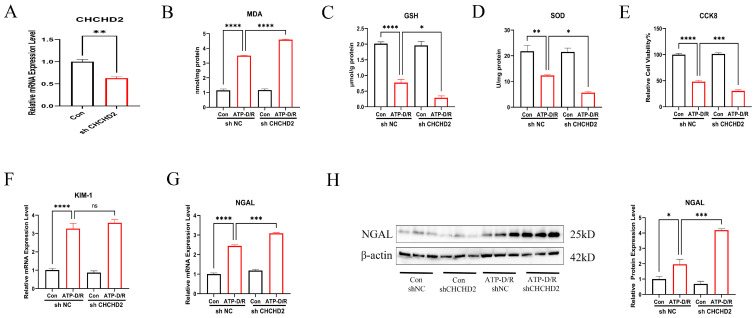
CHCHD2 knockdown exacerbated oxidative stress and cellular injury. (**A**) Validation of CHCHD2 KD was completed. (**B**–**E**) In KD cells, MDA levels were significantly increased, GSH levels and SOD activity were further reduced, and cell viability was decreased as determined by the CCK8 assay. (**F**) Quantitative PCR analysis revealed that CHCHD2 KD elevated KIM-1 RNA expression. (**G**,**H**) CHCHD2 KD significantly increased NGAL RNA and protein expression. Values are presented as the mean ± standard error of the mean (SEM) (n = 3). Statistical significance was determined using a one-way ANOVA followed by Tukey’s multiple comparisons test, with ns indicating no significance, * *p* < 0.05, ** *p* < 0.01, *** *p* < 0.001, and **** *p* < 0.0001.

**Figure 5 ijms-26-06089-f005:**
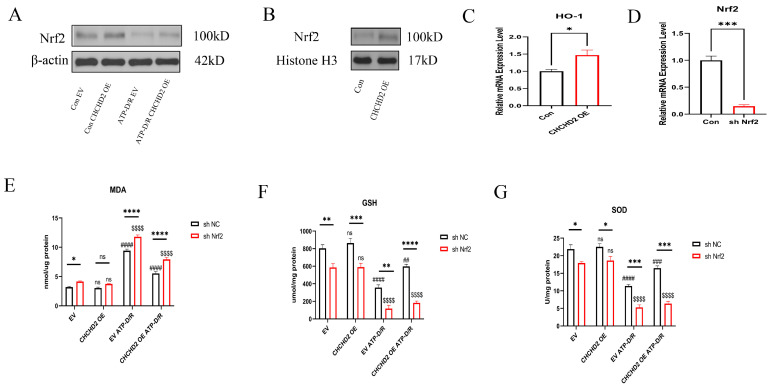
CHCHD2 modulated the Nrf2 signaling pathway to attenuate oxidative stress injury. (**A**) Under ATP-D/R conditions, Nrf2 protein expression was suppressed, while CHCHD2 OE increased Nrf2 expression. (**B**) After CHCHD2 OE, the nuclear level of Nrf2 was elevated. (**C**) CHCHD2 OE upregulated the expression of the downstream antioxidant factor HO-1. (**D**) Nrf2 KD was performed in CHCHD2 OE cells. (**E**) After Nrf2 KD, MDA levels increased in the CHCHD2 OE group. (**F**) After Nrf2 KD, GSH levels were significantly reduced in the CHCHD2 OE group. (**G**) After Nrf2 KD, SOD activity significantly decreased in the CHCHD2 OE group. Values are presented as the mean ± standard error of the mean (SEM) (n = 3). Statistical significance was determined using a two-way ANOVA followed by Šídák’s multiple comparisons test, with ns indicating no significance, * *p* < 0.05, ** *p* < 0.01, *** *p* < 0.001, and **** *p* < 0.0001 for the indicated comparisons. Additionally, two sets of symbols are used for comparisons relative to the EV group: ns indicates no significance, ^##^ *p* indicates < 0.01, ^###^ *p* indicates < 0.001, and ^####^ *p* or ^$$$$^ *p* indicates < 0.0001.

**Table 1 ijms-26-06089-t001:** Primer sequence list.

Genes	Accession Number	Primer (5′→3′) Sequences
β-actin	NM_001101.5	Forward: CAGGATGCAGAAGGAGATCACT Reverse: CGATCCACACGGAGTACTTGC
Kim-1	NM_024446023.2	Forward: CCGTGACAGAGTCTTCAGATGG Reverse: AGCAAGAAGCACCAAGACAGA
NGAL	NM_005564.5	Forward: GAGCACCAACTACAACCAGCA Reverse: TCCTTTAGTTCCGAAGTCAGCT
CHCHD2	NM_001320327.2	Forward: AGGAAGTAATGCTGAGCCTG Reverse: ACCCTCACAGAGCTTGATGTC
HO-1	NM_002133.3	Forward: CTGCTGACCCATGACACCAAG Reverse: CTGTCGCCACCAGAAAGCTGA
Nrf2	NM_001145412.3	Forward: TGTGGCATCACCAGAACACT Reverse: TCCAGGGGCACTATCTAGCTC

## Data Availability

The data presented in this study are available on request from the corresponding author.

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
