# Peer review of "Therapeutic Potential of CHCHD2 in Ischemia–Reperfusion Injury: Mechanistic Insights into Nrf2-Dependent Antioxidant Defense in HK2 Cells"

_ijms, 2025, doi:10.3390/ijms26136089_

Round 1
Reviewer 1 Report
Comments and Suggestions for Authors
All my comments are in the Word file attached.

Author Response
Response to Reviewer 1
Dear Reviewer 1,
We sincerely appreciate the reviewer’s thoughtful and positive comments regarding our manuscript entitled “Therapeutic Potential of CHCHD2 in Ischemia–Reperfusion Injury: Mechanistic Insights into Nrf2-Dependent Antioxidant Defense in HK2 Cell”. We are grateful for the reviewer’s recognition of the scientific rationale, particularly the justification of using a mitochondrial approach to investigate the protective role of CHCHD2 in the ATP depletion/recovery model of acute kidney injury (AKI).
We acknowledge the reviewer’s point regarding the limitation of the study being conducted solely in vitro. As noted by the reviewer, we have clearly stated this limitation in the discussion section of the manuscript and emphasized the necessity of future in vivo validation.
We are particularly encouraged by the reviewer’s conclusion that our findings support the role of CHCHD2 in enhancing Nrf2 signaling and promoting antioxidant gene expression, such as HO-1. We deeply appreciate the reviewer’s recommendation for acceptance and publication.
Thank you again for your valuable feedback.
Sincerely,
Zhou
On behalf of all authors
Reviewer 2 Report
Comments and Suggestions for Authors
In their study entitled “Therapeutic Potential of CHCHD2 in Ischemia–Reperfusion Injury: Mechanistic Insights into Nrf2-Dependent Antioxidant Defense in HK2 Cell” Yajie Hao and Xiaoshuang Zhou describe an in vitro model of ischemia/reperfusion injury and investigate the antioxidant and protective effect of Coiled-coil-helix-coiled-coil-helix domain containing 2 (CHCHD2). The results are clearly described but there are several concerns that need further investigations.
- Figure 2A Under ATP-D/R conditions, CHCHD2 expression was significantly reduced. The legend of histogram graph and the related blot image are different. Please adjust to better clarify that the red histogram refers to the decrease of CHCHD2 overexpression in presence of ATP-D/R
- The analysis of NGAL should be carried out also by western blot to confirm or not the gene expression results
- The discussion should be revised by describing results in the context of the literature available
- Since CCCP disrupts mitochondrial membrane potential, assays that evaluate mitochondrial function should be carried out to strength the data regarding ROS production.
Minor issues: The results paragraph is named as “Conclusion”, please correct
Author Response
Response to Reviewer 2
Dear Reviewer 2,
We sincerely thank Reviewer 2 for the careful evaluation of our manuscript entitled “Therapeutic Potential of CHCHD2 in Ischemia–Reperfusion Injury: Mechanistic Insights into Nrf2-Dependent Antioxidant Defense in HK2 Cells” and for the valuable suggestions that will greatly improve the quality of our work. We have carefully revised the manuscript in response to each of the comments, as detailed below:
- Comment: Figure 2A Under ATP-D/R conditions, CHCHD2 expression was significantly reduced. The legend of histogram graph and the related blot image are different. Please adjust to better clarify that the red histogram refers to the decrease of CHCHD2 overexpression in presence of ATP-D/R.
Response: We thank the reviewer for this important observation. We have revised the legend of Figure 2A and updated the histogram color annotation to clearly indicate that the red histogram represents CHCHD2 expression under ATP-D/R conditions. The blot image and histogram are now fully consistent, and the description has been clarified in the figure legend accordingly.
- Comment: The analysis of NGAL should be carried out also by western blot to confirm or not the gene expression results.
Response: We thank the reviewer for this constructive suggestion. In response, we have performed Western blot analyses to evaluate NGAL protein expression under ATP-D/R conditions in both CHCHD2 overexpression and knockdown models. The results are now presented in Figure 3E (CHCHD2 overexpression) and Figure 4H (CHCHD2 knockdown). These findings are consistent with our qPCR results and further confirm the regulatory role of CHCHD2 on NGAL expression. Additionally, the figure legends have been revised and highlighted in red font to clearly indicate the updated content.
- Comment: The discussion should be revised by describing results in the context of the literature available.
Response: We appreciate the reviewer’s valuable suggestion. In the revised manuscript, we have substantially rewritten the Discussion section to emphasize comparisons between our findings and previously published studies. This revision deepens the interpretation of our results by integrating relevant literature, rather than merely presenting experimental outcomes. The revised sections are marked in red font for clarity.
- Comment: Since CCCP disrupts mitochondrial membrane potential, assays that evaluate mitochondrial function should be carried out to strengthen the data regarding ROS production.
Response: We appreciate this suggestion and agree that evaluating mitochondrial function is essential. In the revised manuscript, we have included JC-1 staining to assess mitochondrial membrane potential (Δψm) under ATP-D/R conditions. The results show a significant loss of Δψm after ATP-D/R, which is partially rescued by CHCHD2 overexpression. These data are now presented in Figure 2G and discussed in the section 2.2.
Minor Issue: The results paragraph is named as “Conclusion”, please correct.
Response: We apologize for the oversight. The section title has been corrected from “Conclusion” to “Results” in the revised manuscript.
We are grateful for your constructive comments, which have significantly improved the clarity and scientific rigor of our manuscript. We hope that the revised version meets your expectations.
All revised and corrected sections have been highlighted in red font for clarity.
Sincerely,
Dr. Zhou
Round 2
Reviewer 2 Report
Comments and Suggestions for Authors
The authors have adequately addressed all the suggestions, significantly improving the results and the discussion accordingly.